# Prevalence and Antibiogram of *Vibrio parahaemolyticus* and *Aeromonas hydrophila* in the Flesh of Nile Tilapia, with Special Reference to Their Virulence Genes Detected Using Multiplex PCR Technique

**DOI:** 10.3390/antibiotics10060654

**Published:** 2021-05-30

**Authors:** Hanan A. Zaher, Mohamad I. Nofal, Basma M. Hendam, Moustafa M. Elshaer, Abdulaziz S. Alothaim, Mostafa M. Eraqi

**Affiliations:** 1Food Hygiene and Control Department, Faculty of Veterinary Medicine, Mansoura University, Mansoura 35516, Egypt; hananzaher@mans.edu.eg; 2General Authority of Fish Resources and Development (GAFRD), Manzala Fish Farm, Manzala 35642, Egypt; aemohamadonofal@yahoo.com; 3Department of Husbandry and Development of Animal Wealth, Faculty of Veterinary Medicine, Mansoura University, Mansoura 35516, Egypt; basmahendam@mans.edu.eg; 4Department of Microbiology at Specialized Medical Hospital, Mansoura University, Mansoura 35516, Egypt; moustafaelshaer@mans.edu.eg; 5Department of Biology, College of Science in Zulfi, Majmaah University, Majmaah 11952, Saudi Arabia; 6Microbiology and Immunology Department, Veterinary Research Division, National Research Centre, Dokki, Giza 12622, Egypt

**Keywords:** molecular identification, *Aeromonas hydrophila*, *Vibrio parahaemolyticus*, *Oreochromis niloticus*

## Abstract

*Vibrio parahaemolyticus* and *Aeromonas hydrophila* are major public health problems and the main cause of bacterial disease in Nile tilapia (*Oreochromis niloticus*). This study was conducted to determine the prevalence, antibiotic resistance and some virulence genes of both *V. parahaemolyticus* and *A. hydrophila* isolates from Nile tilapia. From Manzala Farm at Dakahlia governorate, 250 freshwater fish samples were collected. The confirmed bacterial isolates from the examined Nile tilapia samples in the study were 24.8% (62/250) for *V. parahaemolyticus* and 19.2% (48/250) for *A. hydrophila*. multiplex PCR, revealing that the *tlh* gene was found in 46.7% (29/62) of *V. parahaemolyticus* isolates, while the *tdh* and *trh* virulence genes were found in 17.2% (5/29). Meanwhile, 39.5% (19/48) of *A. hydrophila* isolates had the *16s rRNA* gene and 10.5% (2/19) had the *aerA* and *ahh1* virulence genes. The Multiple Antibiotic Resistance indices of *V. parahaemolyticus* and *A. hydrophila* were 0.587 and 0.586, respectively. In conclusion, alternative non-antibiotic control strategies for bacterial infections in farmed fish should be promoted to avoid multidrug-resistant bacteria. Therefore, it is suggested that farmers should be skilled in basic fish health control and that molecular detection methods are more rapid and cost-effective than bacteriological methods.

## 1. Introduction

Aquaculture is a globally essential industry that provides food to the world’s rapidly growing population, in addition to being a good source of low-cost animal protein [1]. Nile tilapia (*Oreochromis niloticus*) is the second-largest aquatic species cultivated worldwide [2,3]. The overall production of farmed Nile tilapia in Egypt accounts for 71.2 percent of all farmed Nile tilapia worldwide [4] and is the source of the most common strains and species used in commercial aquaculture. In general, seafood is a notable source of pathogenic microorganisms infectious to humans [5,6]. Cultivation of fish may pose some fish health problems due to contamination by pathogenic bacteria, which leads to high economic losses [7]. Contamination can occur at any time during the collection, storage or distribution of seafood and may come from water, facilities, machinery and handlers. Feces containing spoilage microorganisms and pathogens may contaminate seafood and cause microbial contamination [8]. Freshwater tilapia in Egypt is often exposed to a variety of possible stressors, including inadequate management and environmental factors that reduce natural immunity, resulting in disease and death in the fish [9]. The key environmental parameters are increased turbidity, temperature, salinity, pH, water conductivity and lower dissolved oxygen [4]. Tilapia, under stress factors, is susceptible to many bacterial diseases that are common in the freshwater ecosystem [10]. Bacteria are responsible for huge mortalities in fish species resulting in higher economical loss in aquaculture [11]. However, bacteria are not the only main cause of infection in fish; viruses also have a major role in aquaculture contamination. The major viral fish diseases in aquaculture have been reported in many reviews [12,13,14,15,16].

*V. parahaemolyticus* is out of the most common causes of salty, brackish and freshwater fish diseases [17]. The usual clinical signs of the *V. parahaemolyticus* toxin in humans include acute dysentery and stomach ache, followed by diarrhea, nausea, vomiting, fever, chills and watery stool [18]. The genes responsible for haemolysin, comprise thermostable direct-haemolysin (*tdh*), TDH-related haemolysin (*trh*) and thermolabile haemolysin (*tlh*), performs an essential role in pathogenesis [19,20]. In this regard, *V. parahaemolyticus* strains possess some virulence factors as *tdh* and *trh* [21], that are linked mainly to hemolysis and cytotoxicity in the host cell [22]. This means that most *V. parahaemolyticus* found in clinical cases carry *trh* or *tdh* genes [23]. Detection of the presence of *Vibrio* in a food sample is usually performed by selective media like thiosulphate citrate bile salts sucrose (TCBS) media, and then confirmed by biochemical assays [24,25]. *Aeromonas* spp. causes more severe bacterial diseases that influences various fish and shellfish species, posing a serious threat to Egypt’s and other countries’ fish farming industries [26]. *A. hydrophila*, on the other hand, is a zoonotic pathogen belonging to the Aeromonadaceae family [27,28,29,30], which is considered the most important cause of gill and skin disease, causing high mortality rate in freshwater fish [31]. It may cause intestinal and extra-intestinal diseases in humans, such as septic arthritis, diarrhea (traveler’s diarrhea), gastroenteritis, meningitis, septicemia and skin and wound infections [32]. Virulence factors like DNases, hemolysin, proteases, aerolysin and lipases has been corelated with the pathogenicity of *Aeromonas* spp. Such toxins play a critical role in the development of diseases in fish and humans [33,34]. *16S rRNA* is a suitable marker for *Aeromonas* spp. identification [35]. Antibiotics are widely utilized in seafood farms as feeding additives or to prevent bacterial diseases [36], but extreme usage of antibiotics resulted in the evolution of drug resistance in aquaculture pathogenic strains, which has become a serious problem for veterinary and human health [37]. The Multiple Antibiotic Resistance (MAR) Index is considered as a useful way to evaluate contamination sources [38]. This study aimed to detect the prevalence and antibiotic-resistant patterns of *V. parahaemolyticus* and *A. hydrophila* in Nile tilapia samples collected from Manzala Farm at Dakhalia governorate. Moreover, the presence of certain virulence factors was investigated using the multiplex PCR technique for detecting the occurrence of this important seafood-borne pathogen and providing a fortuitous opportunity to high-risk environments and consequently reducing the risk of food-borne illness.

## 2. Results

### 2.1. The Environmental Parameter at Manzala Farm

These data were taken from the farm administrator: Oxygen = 3 mg/L, PH = 9.1, Temp = 33 °C, Salinity 1.3 g/L, NH3 = over range (more than 1 mg/L), NH4 = over range (more than 1 mg/L), Copper = 1.2 mg/L, Hardness = more than 500 mg/L, Iron = 1.3 mg/L, Nickel = 1.55 mg/L, Phosphates = 2 mg/L, Sulfide = 0.4 mg/L, Nitrite = 0.75mg/L, Zinc = 0.3 mg/L. All these environmental parameters are above the permissible levels. The mortality rate was about 10% because of poor storage in ponds and the use of agricultural drainage water as a resource for pond water.

### 2.2. Clinical Signs and Postmortem Examination of Diseased Tilapia

Postmortem examination showed a congested liver with hemorrhage on its surface and engorged gall bladder with splenic hemorrhage. The kidney was also congested and enlarged in the case of *V. parahaemolyticus* (Figure 1a,b), while the clinical signs of naturally infected fish with Aeromonas were hemorrhagic septicemia in the form of bilateral exophthalmia with gill cover hemorrhage, eye clouding, hemorrhage, surface ulcers, abdominal distension and massive mortality (Figure 1c,d).

### 2.3. Prevalence of Comfirmed Isolates of V. parahaemolyticus and A. hydrophila in the Examined Nile tilapia Based on Biochemical Tests

In this study, the detectable prevalence rates of confirmed isolates of *V. parahaemolyticus* and *A. hydrophila* were 24.8% (62/250) and 19.2% (48/250), respectively (Table 1; Figure 2).

### 2.4. Molecular Identification:

#### 2.4.1. Molecular identification of *V. parahaemolyticus*

Multiplex PCR was performed on the *V. parahaemolyticus* isolates. It was found that only 29 (46.7%) of the samples contained the thermolabile hemolysin (*tlh*) gene at 450 bp, and only 5 (17.2%) of the 29 samples contained both the thermostable direct hemolysin (*tdh*) gene at 269 bp and *tdh*-related hemolysin (*trh*) gene at 500 bp (Table 1; Figure 2 and Figure 3).

#### 2.4.2. Molecular identification of *A. hydrophila*

Multiplex PCR was done on the *A. hydrophila* isolates; to distinguish the presence of virulence genes, it was found that only 19 (39.5%) samples contained the *16S rRNA* gene at 356 bp, and only 2 (10.5%) from 19 samples contained the aerolysin (*aerA*) gene at 309 bp and extracellular hemolysin (*ahh1*) gene at 130 bp (Table 1; Figure 2 and Figure 4).

### 2.5. Antibiotic Susceptibility of V. parahaemolyticus and A. hydrophila Positive Strains Confirmed by Multiplex PCR Assay

An antibiogram sensitivity test was performed on the 29 positive samples of the *V. parahaemolyticus* strains, which were confirmed by multiplex PCR. This indicated that *V. parahaemolyticus* was highly sensitive to ampicillin and amikacin (65.7%). The intermediate was exhibited against neomycin (48.4%), cefotaxime (34.4%), sulfamethoxazole, tetracycline and ciprofloxacin (31.0%). However, a higher resistance pattern varied among the other tested drugs; the highest resistance (100%) was recorded for cloxacillin, streptomycin and erythromycin, followed by nalidixic acid (82.7%) (Table 4; Figure 5). On the other hand, *A. hydrophila* strains were highly sensitive to gentamycin (84.2%), tetracycline, ciprofloxacin (68.4%), amikacin, neomycin and kanamycin (52.6%), and demonstrated high resistance to cloxacillin and erythromycin (100%), followed by cefotaxime and streptomycin (84.2%), nalidixic acid, sulfamethoxazole and cephalothin (68.4 %) (Table 2; Figure 6).

The MAR index values showed multiple resistant patterns, revealing that the MAR index averages of *V. parahaemolyticus* and *A. hydrophila* were 0.587 (Table 3) and 0.586 (Table 4), respectively. 

## 3. Discussion

Hadous drainage is the primary water supply for Manzala fish farm, but unfortunately, it is contaminated by agricultural and industrial activities, especially during rice cultivation in the summer due to the use of many chemicals, and during canal cleaning in the winter. Both factors contribute to the pollution of Hadous water sources, putting fish and their development under stress [39].

Another issue is the abundance of *Eichhornia* spp., which consumes large quantities of water containing all the nutrients needed for fish production, leaving the water devoid of basic plankton species. When these plants die, they decay, producing toxic gasses such as hydrogen disulfide, as well as some trace elements like copper, iron, phosphorus, manganese, and nitrogenous compounds, that increase stress on the fish in this farm. Finally, the fluctuating water level of Hadous drainage throughout the year reflects Manzala fish-farm water level, contributing to a reduction in the water quality and increasing stress on the fish in this farm [40].

As a result of the length of the breeding period (about 18 months) and the increase in the organic matter at the bottom of the pond, there has been a severe decrease in dissolved oxygen in the water, with an increase in the pH and toxic ammonia in the water that plays important roles in the multiplication of pathogens leading to diseases of fishes [41,42,43].

Typical signs of septicemia, such as deep dermal ulceration, abdominal dropsy, skin roughness, skin abrasion and protruded hemorrhagic anal opening, have been shown in the diseased tilapia according to the current research. Postmortem lesions have also been evident. Furthermore, spleen and liver congestion have been determined. These results are in agreement with the previously reported work [44,45].

The prevalence of *Vibrio* was detected by [46] who reported a lower incidence of *Vibrio* spp. in cultured Nile tilapia (16.48 %) and a higher prevalence of 98.67% in freshwater fish [47]. However, other studies reported a comparable prevalence to this study, for example, a lower prevalence of *A. hydrophila* (13.2 %) in the isolated Nile tilapia [48] and as the most prevalent bacterial pathogens (43%) in Nile tilapia in Uganda [49].

An MAR index above 0.2 is a parameter used to reveal the spread of bacterial resistance in certain population. An MAR index more than 0.2 reveals that the strains of such bacteria originated from a habitat in which different antibiotics are used and the abuse of antibiotics in aquatic and environmental systems [50,51]. A high incidence of MAR is permitted by genetic exchange between MAR pathogens and other bacteria [38]. In this study, the MAR indices of *V. parahaemolyticus* and *A. hydrophila* isolates were 0.587 and 0.586, respectively, revealing that these isolates were derived from samples from high-risk sources [52]. This is not surprising because the *V. parahaemolyticus* has shown 100% resistance to tested antibiotics in this research such as streptomycin, erythromycin and cloxacillin. This finding corroborates the ideas of various researchers [52,53,54,55,56] who discovered that several *V. parahaemolyticus* isolates from seafood are resistant to multiple antibiotics. Another study [56] reported that *V. parahaemolyticus* recovered from commonly consumed aquatic products in Shanghai was 75.4% resistant to streptomycin. In contrast, the *A. hydrophila* isolates showed multiple resistance to erythromycin, cloxacillin (100%), streptomycin and cefotaxime (84%). These findings are consistent with a past study [57], which found *A. hydrophila* isolates from red hybrid tilapia in Malaysia were multidrug resistant to cefotaxime, sulfamethoxazole, erythromycin and streptomycin, with an MAR index of 0.5. Similarly, one study [58] found that the MAR index of *A. hydrophila* was between 0.12 to 0.59 and another study [59] found that the MAR index was between 0.243 and 0.457 for Aeromonads isolated from decorative fish farming systems. At the same time, gentamycin was reported as the most active drug against *A. hydrophila* (84%), followed by tetracycline and ciprofloxacin (68.4%); this result is inconsistent with previous findings [60], which reported the lowest sensitivity to gentamycin (40%) and high resistance to chloramphenicol and ciprofloxacin (72% and 48%, respectively). Another study reported that *A. hydrophila* was highly sensitive to ciprofloxacin and resistant to tetracycline [61]. In contrast, *V. parahaemolyticus* was sensitive to ampicillin, but these results disagree with those [62] to have shown that reduced resistance to ampicillin is only found in V*. parahaemolyticus.* Differences in sample sources can be explained by changes in the MAR index [63,64]; antibiotic resistance levels are subject to various selective pressures due to geographic spread [52] and methodologies [65].

The use of antimicrobials in aquaculture, could impose an impact on the development of resistance in human health known as a direct spread of resistance from aquatic environments to human. Similarly, increasing microbial resistance problems could spread from country to another, for instant, the export/import of foods like fish and fishery products [51].

The risk of *V. parahaemolyticus* in seafood to human health was determined by detecting the microorganisms, followed by PCR-based detection of the genes that generate the *tdh* and *trh* toxins. About 2.5 percent of 120 seafood samples tested positive for one or both virulent genes *tdh* and *trh*, according to another major study [66]. Although one study found that shellfish samples tested in Chile were more likely to be positive for *tdh*, with 85 percent (17/20) of overall tested samples being positive [67]. Furthermore, the *tdh* and *trh* genes were found in 8.16 and 12.24% of fish samples South China, respectively [68]. One study [69] contradicts our results; they approved the occurrence of *V. parahaemolyticus* based on PCR with a lower frequency than ours, and about 3% of collected seafood samples were positive for *tdh* and *trh* virulence genes. Other studies were conducted [70,71] that approved the occurrence of *V. parahaemolyticus* at higher frequencies. Various researchers isolated *V. parahaemolyticus* from Thai shrimp samples and discovered that these isolates lacked the *tdh* and *trh* virulence genes [72,73,74].

Manzala Lake is furious with nutrients such as calcium, magnesium, and chlorides; that are closely related to the number of *Aeromonas* spp. found in brackish water [40]*. A. hydrophila* has gained increased attention because of its pathogenicity to humans and the widespread prevalence of the organism in the environment, food and water [75]. Traditional microbiological procedures of *Aeromonas* spp. from food samples are time consuming. The polymerase chain reaction technique has been developed to solve these problems [76,77], and the incidence of *A. hydrophila* isolated from tilapia investigated. It secretes multiple virulence parameters, such as aerolysin and α-hemolysin [78]. The detection of the *16S rRNA* gene has contributed to the rapid and accurate characterization of the bacteria [22]. In addition to biochemical tests, the *16S rRNA* gene is an essential tool in diagnostic laboratories for recognizing microbes [79]. In this regard, approximately 39.5% of these isolates were positive for the *16srRNA* gene and 10.5% were positive for the *aerA* and *ahh1* genes. In addition, the *aerA* and *ahh1* virulence genes were present in *A. hydrophila*, which was confirmed by applying PCR with a prevalence of approximately 52.6% [80]. Wang et al. performed multiplex PCR for the recognition of the *aerA* and *ahh1* genes in *A. hydrophila* and *A. sobria* [77]. The virulence range of aeromonads may originate from variations in the genotypes and phenotypes that are found in the environment. The *A. hydrophila* β-hemolysin virulence gene has been isolated from freshwater fish in China [81]. Cloned β-hemolysin sequences have been used to detect pathogenic *A. hydrophila* strains [82]. Our findings are consistent with past research [83], which detected the presence of the *aerA* gene in 85% of pathogenic *A. hydrophila* isolates from fish and pond water, which caused hemolysis of red blood cells, leading to hemorrhagic signs on fish skin and internal organs due to the presence of hemolysin and enterotoxigenic properties [75,84]. A negative PCR result does not prove the absence of the virulence gene, but may result from sequence differences in the primer binding sites [85]. These findings indicate the importance of performing biological tests for determining the virulence factors of some strains and identifying the potential pathogenicity of *A. hydrophila* due to their possible public health risks [86]. [37] detected the prevalence of *ahh1* and *aerA* virulence factors with 28% and 68% frequencies of *A. hydrophila* isolated from fish collected from Damietta governorate in Egypt. The hemolysin gene was detected in our study with a percentage that is lower than that reported in previous studies (30–100%) [87], while the aerolysin gene was identified with a frequency of 68% of the isolates, which is lower than the detected rates that ranged from 70% to 100% in other studies [88,89]. However, another study reported a lower rate of 66.7% [90]. For instance, in another study, the prevalence of the *aerA* gene was high (83.3%) and that of the *ahh1* gene was low (16.7%) [91].

## 4. Materials and Methods

### 4.1. Farm Information

Manzala lake is an important Egyptian lake according to its size and economic value [42]. The study area (Manzala fish-farm) includes fifty two ponds in four sections each consists of thirteen acres, so the total water area production equal about six hundred seventy six acre. The main water source of this farm is Hadous drainage, that is considered as the main drainage to Manzala Lake (49% of the drainage from the eastern delta). This farm is representative of the thousands of farms in Egypt. The volume of production of this farm is about 1400 tons/year.

### 4.2. Fish Sampling

Specimens (n. 250) of Nile tilapia fish were collected from five ponds with constant environment parameters in all ponds for two weeks. 

The fish (Age of fish = 15 months, Length of fish = 15 cm, Weigh of fish = 200–250 gm) collected from the pond had a breeding time of around 18 months with high organic matter at the bottom of the pond, and low dissolved oxygen and high pH and toxic ammonia in the water. The mortality rate was about 10%.

The experiments were done according to the Research Ethics Committee of the Faculty of Veterinary Medicine, Mansoura. The samples were collected in sterilized sealed plastic containers, transported to the laboratory in a cold box below 4 °C, and analyzed immediately [92].

### 4.3. Bacteriological Examination

#### 4.3.1. Preparation of Fish Samples

Five grams of individual fish muscle, under aseptic conditions, was homogenized into 45 mL of 3% NaCl plus 1% alkaline peptone water for *V. parahaemolyticus* enrichment and tryptic soya broth for A. enrichment [93], followed by incubation at 37 °C for 18 h. for *V. parahaemolyticus* and at 37 °C for 24 h. for *A. hydrophila* [94].

#### 4.3.2. Culture Characters

A loop of prepared fish samples was inoculated with TCBS agar was then incubated at 37 °C for 18–24 h [95]. The isolated colonies appeared as green or blue-green colonies (sucrose negative) according to the power to ferment sucrose [96]. Suspected colonies were collected and transferred onto a tryptic soya agar slant enriched with 2% NaCl for further microscopic and biochemical identification. For the isolation of *A. hydrophila*, Aeromonas agar base medium (Rayan) supplemented with ampicillin (5 mg/L) (Oxoid) was used. A loopful from previously incubated enriched samples of fish was inoculated on an Aeromonas agar base containing ampicillin (Oxoid) and incubated at 37 °C for 18–24 h. The isolated colonies appeared green with black centers. Suspected colonies were picked and transferred onto tryptic soya agar slants for further microscopic and biochemical identification [97].

#### 4.3.3. Biochemical Examination

Biochemical identification of bacterial isolates was performed using the method described in Bergey’s Manual^®^ of Systematic Bacteriology [98].

The biochemical tests used were cytochrome-oxidase (Oxoid, Denver, USA), catalase (Al-Goumhoria Co, Cairo, Egypt), oxidation-fermentation medium (O-F) (BioMérieux, Marcy-l’Étoile, France), glucose gas production and indole tests (Al-Goumhoria Co, Cairo, Egypt), Esculin hydrolysis (bile esculin agar medium (DifcoTM, California, USA), Voges-Proskauer tests, arabinose, sucrose, lactose and mannose acid production, lysine decarboxylase and arginine dihydrolase and reduction of nitrates. Further identification was achieved using an analytical profile index (API 20 E system (BioMérieux, Paris, France), as instructed by the manufacturer.

### 4.4. Antimicrobial Susceptibility Testing and MAR Index Value

To verify the sensitivity of the test, 14 antimicrobials were used (Table 5). The single diffusion method was used to evaluate the antimicrobial susceptibility according to [99] for *V. parahaemolyticus* and [100] for *A. hydrophila*, and the results were applied according to [101]. The multiple antibiotic resistance (MAR) index for each strain was done according to the equation stipulated by [83,102,103] as follows: MAR index = number of resistance/total number of antibiotic.

### 4.5. Molecular Identification of V. parahaemolyticus and A. hydrophila Virulence Genes

#### 4.5.1. Molecular Identification of *A. hydrophila* Virulence Genes

Genomic DNA was extracted from *A. hydrophila* isolates using a DNA extraction kit (DNeasy kit, Qiagen, USA) following the manufacturer’s guidelines. Multiplex PCR was performed to detect virulence factors including *16S rRNA*, *aerA* and ahh1 of the *A. hydrophila* isolate. The primer sequences and PCR products are illustrated in Table 6. Each PCR reaction was performed in a total volume of 25 μL containing 12.5 μL of dreamTaq master mix (Green PCR Master Mix (2X), Thermo Scientific), 1 μL of each primer and 5 μL of DNA template, and the total volume was completed to 25 μL using DNase/RNase-free water. The PCR thermal conditions are illustrated in Table 6. The first step was denaturation at 94 °C for 4 min, followed by 35 cycles of denaturation at 94 °C for 30 s, annealing for 30 s at the specified temperature according to each gene (57 °C for *tlh*, *tdh* and *trh*; 59 °C for 16S rRNA, ahh1 and *aerA*) and an extension step at 72 °C for 30 s. After the end of the cycles, a final extension step at 72 °C for 10 min was added. The integrity of the PCR products was checked by electrophoresis on 1.5% agarose gels and visualized with a UV trans-illuminator with a 100-bp DNA ladder (Invitrogen, San Jose, CA, USA), which was used as the size standard.

#### 4.5.2. Molecular Identification of *V. parahaemolyticus* Virulence Genes

The amplification was done on a Thermal Cycler (Master Cycler, Eppendorf, Hamburg, Germany). PCR was done in a 25 mL volume consisting of 0.5 mg of genomic DNA, 0.5 mM of each of the oligonucleotide primers for *tlh, tdh* and *trh* (1.25 mL of each of the primers from a 20 mM stock suspension), 2.5 mL of a 10_PCR reaction buffer (500 mM Tris-Cl, pH 8.9, 500 mM KCl and 25 mM MgCl2), 0.5 mL 10 mM dNTPs, 1.25 units Taq DNA polymerase and an appropriate volume of sterile MilliQ water. Amplification of the DNA segment was performed with the following temperature cycling parameters: initial denaturation at 95 °C for 5 min followed by 30 cycles of amplification. Each cycle consisted of denaturation at 95 °C for 1 min, primer annealing at 58 °C for 1 min, primer extension at 72 °C for 1 min and a final extension at 72 °C for 10 min. Of each amplified product, 10 µL was separated in 1.5% agarose gel by electrophoresis. The gel was then stained with ethidium bromide (0.5 mg/mL) and visualized on a UV transilluminator. A 100 bp plus DNA Ladder was used to determine the fragment sizes.

## 5. Conclusions

Aquaculture in Egypt continues to be a growing, vital and important high-protein, easily digestible and high-value production sector for livestock. However, disease outbreaks are a major problem in aquaculture. Environmentally transmitted bacterial pathogens in their hosts can result in single and combined co-infections that have significant economic impacts on the aquaculture of Egypt. Antibiotic-resistant *V. parahaemolyticus* and *A. hydrophila* isolates in seafood may pose a danger to human health, and according to this study, adequate control measures should be implemented to reduce the risk of contamination and avoid antibiotic resistance. This would reduce the risk of transferring antibiotic-resistant bacteria to the human population through fish products. Therefore, antibiotic susceptibility should be determined in broader studies. It is important to promote a focus on alternative non-antibiotic control strategies for bacterial infections in farmed fish. To fully comprehend bacterial effects on Egyptian fish and human health, characterization of the isolated bacteria, including pathogenicity studies, is important. Future research should also concentrate on the use of more accurate methods of bacterial identification in order to identify contaminated seafood.

## Figures and Tables

**Figure 1 antibiotics-10-00654-f001:**
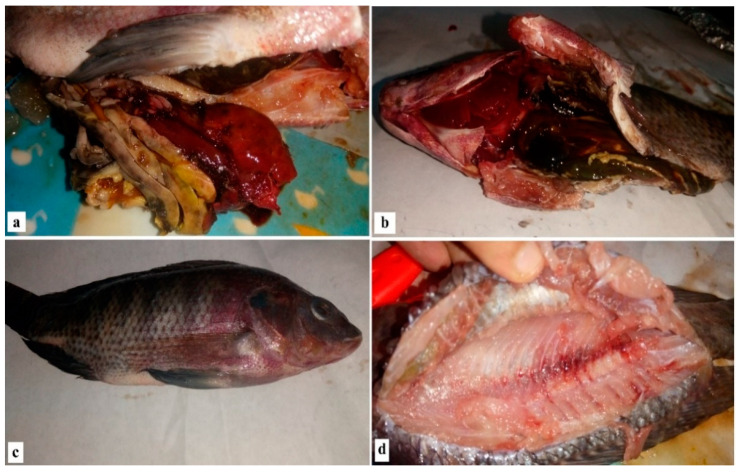
Clinical signs and postmortem examination of diseased Nile tilapia.

**Figure 2 antibiotics-10-00654-f002:**
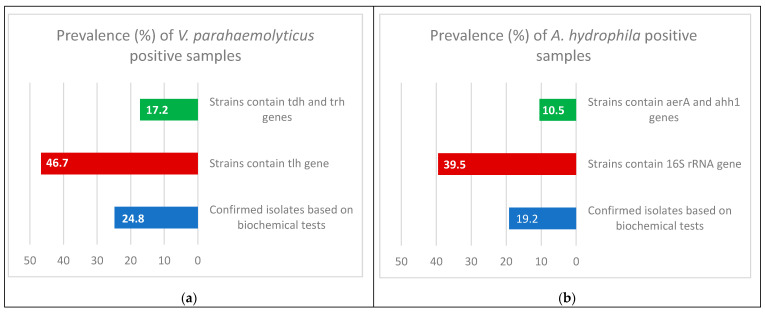
Prevalence and molecular identification of confirmed isolates of *V. parahaemolyticus* (**a**) and *A. hydrophila* (**b**) in Nile tilapia.

**Figure 3 antibiotics-10-00654-f003:**
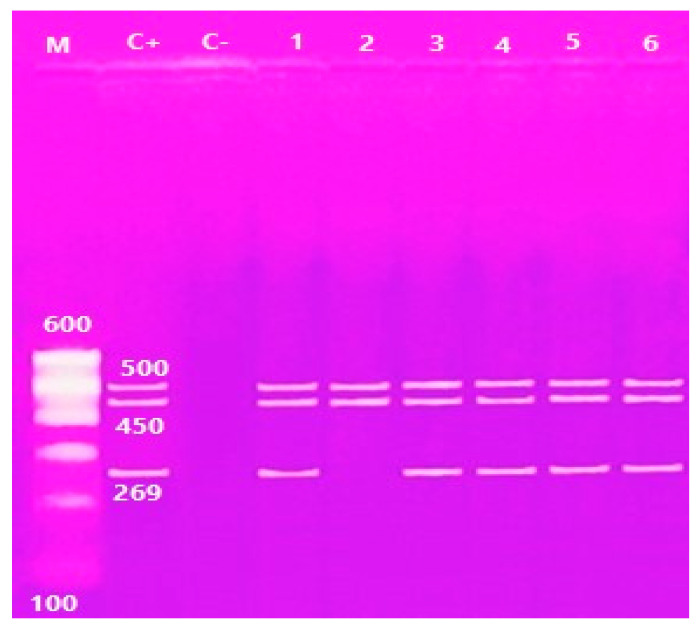
Agarose gel electrophoresis of multiplex PCR of *trh* (500 bp) *tlh* (450 bp) and *tdh* (269 bp) genes of *V. parahaemolyticus*. Lane M: 100 bp ladder, Lane C+: control positive, Lane C-: Control negative; Lanes 1, 2, 3, 4, 5 and 6: positive strains for *trh* genes; Lane 1, 2, 3, 4, 5 and 6: positive strains for *tlh* genes; Lanes 1, 3, 4, 5 and 6: positive strains for *tdh* genes.

**Figure 4 antibiotics-10-00654-f004:**
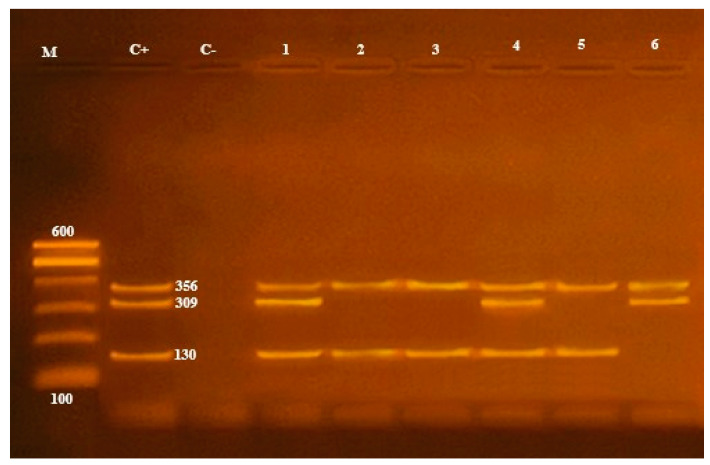
*16S rRNA* (356 bp), *aerA* (309 bp) and *ahhl* (130 bp) virulence genes to characterize *A. hydrophila*. Lane M: 100 bp ladder, Lane C+: control positive, Lane C-: control negative; Lanes 1, 2, 3, 4, 5 and 6: positive strains for *16S rRNA* genes; Lanes 1, 4 and 6: positive strains for *aerA* genes; Lanes 1, 2, 3, 4 and 5: positive strains for *ahhl* genes.

**Figure 5 antibiotics-10-00654-f005:**
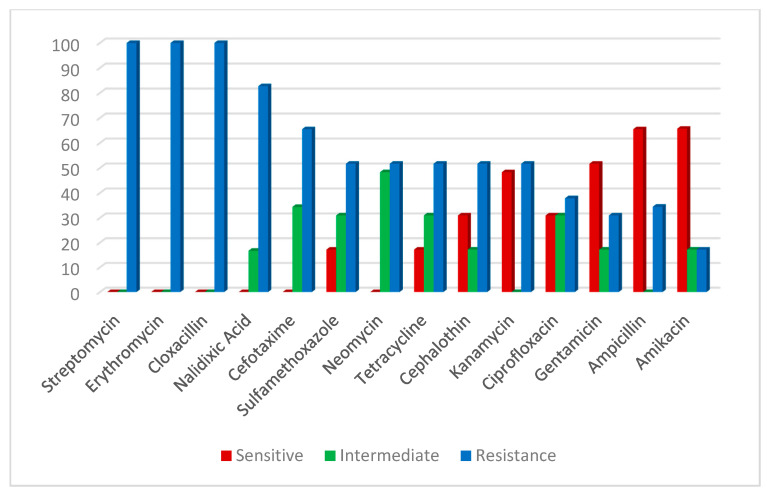
Antibiotic susceptibility of *V. parahaemolyticus*.

**Figure 6 antibiotics-10-00654-f006:**
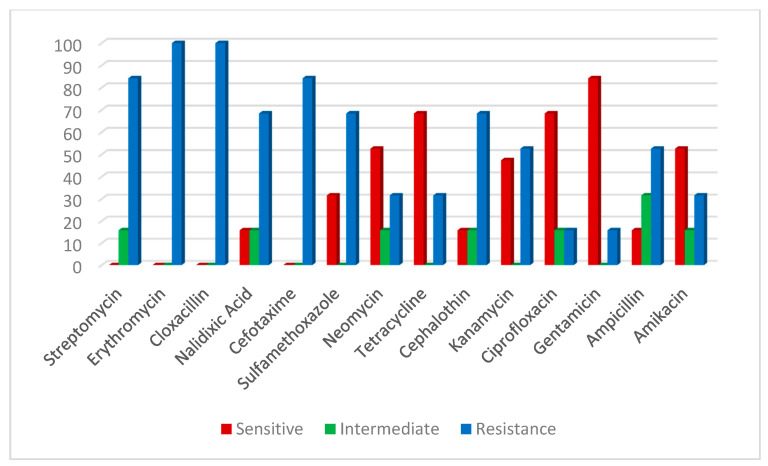
Antibiotic susceptibility of *A. hydrophila*.

**Table 1 antibiotics-10-00654-t001:** Prevalence and molecular identification of confirmed isolates of *V. parahaemolyticus* and *A. hydrophila* in Nile tilapia (*n** = 250).

Isolates	Confirmed Isolates		Strains Containing *tlh* Gene		Strains Containing *tdh* and *trh* Genes	
	*n*	%	*n*	%	*n*	%
*V. parahaemolyticus*	62	24.8	29	46.7	5	17.2
*A. hydrophila*			Strains contain *16S rRNA* gene		Strains contain *aerA* and *ahh1* genes	
	48	19.2	19	39.5	2	10.5

*n**: Number of total samples.

**Table 2 antibiotics-10-00654-t002:** Antibiotic susceptibility of *V. parahaemolyticus* (*n* = 29) and *A. hydrophila* (*n* = 19).

Antimicrobial Agent	*V. parahaemolyticus*						*A. hydrophila*					
	Sensitive		Intermediate		Resistant		Sensitive		Intermediate		Resistant	
	*n*	%	*n*	%	*n*	%	*n*	%	*n*	%	*n*	%
Streptomycin	0	0	0	0	29	100	0	0	3	15.8	16	84.2
Erythromycin	0	0	0	0	29	100	0	0	0	0	19	100
Cloxacillin	0	0	0	0	29	100	0	0	0	0	19	100
Nalidixic Acid	0	0	5	16.7	24	82.7	3	15.8	3	15.8	13	68.4
Cefotaxime	0	0	10	34.4	19	65.5	0	0	0	0	16	84.2
Sulfamethoxazole	5	17.2	9	31.0	15	51.7	6	31.6	0	0	13	68.4
Neomycin	0	0	14	48.3	15	51.7	10	52.6	3	15.8	6	31.6
Tetracycline	5	17.2	9	31.0	15	51.7	13	68.4	0	0	6	31.6
Cephalothin	9	31.0	5	17.2	15	51.7	3	15.8	3	15.8	13	68.4
Kanamycin	14	48.3	0	0	15	51.7	9	47.4	0	0	10	52.6
Ciprofloxacin	9	31.0	9	31.0	11	37.9	13	68.4	3	15.8	3	15.8
Gentamicin	15	51.7	5	17.2	9	31.0	16	84.2	0	0	3	15.8
Ampicillin	19	65.5	0	0	10	34.5	3	15.8	6	31.6	10	52.6
Amikacin	19	65.7	5	17.2	5	17.2	10	52.6	3	15.8	6	31.6

*n*: Number of tested positive bacterial strains confirmed by multiplex PCR assay.

**Table 3 antibiotics-10-00654-t003:** Antibiotics resistance profile and MAR index of *V. parahaemolyticus* isolated from Nile tilapia (*n** = 29).

No. of Tested *V. parahaemolyticus* Strains	Antimicrobial Resistance Profile	MAR Index
5	S, E, CL, NA, CF, SXT, N, T, CN, K, CP, G, AM, AK	1
4	S, E, CL, NA, CF, SXT, N, T, CN, K, CP, G, AM	0.928
1	S, E, CL, NA, CF, SXT, N, T, CN, K, CP, AM	0.857
1	S, E, CL, NA, CF, SXT, N, T, CN, K, CP	0.785
4	S, E, CL, NA, CF, SXT, N, T, CN, K	0.714
4	S, E, CL, NA, CF	0.357
5	S, E, CL, NA	0.285
4	S, E, CL	0.21
1	S, E	0.142
Average		0.587

*n**: total number of *V. parahaemolyticus* positive samples confirmed by multiplex PCR assay.

**Table 4 antibiotics-10-00654-t004:** Antibiotic resistance profile and MAR index of *A. hydrophila* isolated from Nile tilapia (*n** = 19).

No. of Tested *A. hydrophila* Strains	Antimicrobial Resistance Profile	MAR Index
3	E, CL, S, CF, NA, SXT, CN, K, AM, *n*, T, AK, CP, G	1
3	E, CL, S, CF, NA, SXT, CN, K, AM, N, T, AK	0.857
4	E, CL, S, CF, NA, SXT, CN, K, AM	0.642
3	E, CL, S, CF, NA, SXT, CN	0.5
3	E, CL, S, CF	0.285
3	E, CL	0.214
Average		0.586

*n**: total number of *A. hydrophila* positive samples confirmed by multiplex PCR assay.

**Table 5 antibiotics-10-00654-t005:** Antimicrobial discs used for *V. parahaemolyticus* and *A. hydrophila.*

Antibiotic	Symbol	Concentration
Streptomycin	S	10 μg
Erythromycin	E	15 μg
Cloxacillin	CL	5 μg
Nalidixic acid	NA	30 μg
Cefotaxime	CF	30 μg
Sulfamethoxazole	SXT	25 μg
Neomycin	N	30 μg
Tetracycline	T	30 μg
Cephalothin	CN	30 μg
Kanamycin	K	30 μg
Ciprofloxacin	CP	5 μg
Gentamycin	G	10 μg
Ampicillin	AM	10 μg
Amikacin	AK	25 μg

**Table 6 antibiotics-10-00654-t006:** Primer sets for PCR amplification of the target genes specific for *V. parahaemolyticus* and *A. hydrophila*.

Target Gene	Oligonucleotide Sequence (5′ → 3′)	Product Size (bp)	References
*V. parahaemolyticus*			
*tlh*	F-5′AAAGCGGATTATGCAGAAGCACTG-′3 R-5′ GCTACTTTCTAGCATTTTCTCTGC-′3	450	[104]
*tdh*	F-**5**′ GTAAAGGTCTCTGACTTTTGGAC ′**3** R-**5**′ TGGAATAGAACCTTCATCTTCACC ′**3**	269	
*trh*	F-**5**′ TTGGCTTCGATATTTTCAGTATCT ′**3** R-**5**′ CATAACAAACATATGCCCATTTCCG ′**3**	500	
*A. hydrophila*			
*16S rRNA*	F-5′ GGGAGTGCCTTCGGGAATCAGA-′3 R-5′ TCACCGCAACATTCTGATTTG-′3	356	[105]
*aerA*	F-5′CAAGAACAAGTTCAAGTGGCCA-′3 R-5′ ACGAAGGTGTGGTTCCAGT-′3	309	
*ahh1*	F-5′ GCCGAGCGCCCAGAAGGTGAGTT-′3 R-5′ GAGCGGCTGGATGCGGTTGT -′3	130	

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
