# Peer review of "Prevalence and Antibiogram of Vibrio parahaemolyticus and Aeromonas hydrophila in the Flesh of Nile Tilapia, with Special Reference to Their Virulence Genes Detected Using Multiplex PCR Technique"

_antibiotics, 2021, doi:10.3390/antibiotics10060654_

Round 1
Reviewer 1 Report
Very interesting data in antibiotic susceptibility test ! However, I am not convinced it is correct to talk write ‘resistance’ here – resistance suggests a genetic change, and this has not been shown. Tolerance could be a better term.
Mention age, length and weight of fish sample
Before analysis of fish tissue, whether it was kept in cooling condition. If yes, please mention in materials and methods
Minor comments
Scientific names should be in italics
Author Response
Thank you for the revision of our article and the evaluation report of the referees. On behalf of the coauthors, I would like to thank the reviewers for their feedback and constructive comments.

Reviewer 2 Report
This manuscript is interesting because it provides prevalence data on Vibrio parahaemolyticus and Aeromonas hydrophila in cultured fish Tilapia in Egypt. However, there are some issues which should be addressed before this paper could be published.
Specific comments:
L21 “additionally based on biochemical tests” this should be the real prevalence, prevalence based only morphology is not sufficient to determine the species
L23-25: The whole sentence needs to be rephrased.
L43: “health problems” +> for who: fish or humans?
L49: “cause disease and death” : again need to distinguish between fish disease and human disease
L49: “physicochemical elements” should be replaced with environmental parameters
L52-53: what about viral diseases?
L54: “that live”: Vibrio or fish?
L55 -56: This part needs a better structure, the previous sentence was about fish disease, this one is about symptoms in humans, but this is not clearly mentioned.
L57: “variation in altered virulence factors”: what does it mean?
L58: should be mentioned the names of the proteins the tdh and trh genes are coding for, and these genes should be in italics.
L60: replace “have carried” with carry
L61: need full name for TCBS
L62-63: meaning?
L67: to add “and” between gill and skin
L70: remove “a group of”, unnecessary
L72-73: unnecessary repetition
L74-75: this needs to be rephrased
L75-78: needs to be rephrased
L78: full name for “MAR index” needed
L79: replace “frequency” with prevalence
L81-84: virulence factors are not investigated by phenotypic methods (not here), and there is also no relation here the antibiotic resistance => please rephrase
L86: need plural for “sign”
L86: the order of the sections needs to be reconsidered as it is only in 2.2, that we know how many fish have been examined, the mortality rate, etc
L97: “morphology” of what?
L98: Were some fish co-infected with both Vp and A. hydrophila? How many?
L99: as the study is focused on specific species of Vibrio and Aeromonas (parahaemolyticus and hydrophila species), the prevalence to be taken into account should only be the prevalence confirmed with the biochemical tests. Colonies identified on the plates are only presumed.
Table 1: “virulent strains contain tlh gene » : tlh gene is not a virulence factor, it is only a marker of parahaemolyticus species. Please correct.
Similar with”virulent strains contain 16SrRNA gene”
“Virulent strains contain tdh and trh”: does it mean here both tdh and trh? What about strains containing only tdh or only trh?
The percentages should also relate to the initial 250 fish being tested.
Figure3: Lanes 1, 3 and 6 show some positive bands for tdh and trh genes, but no band for tlh gene. This is not normal as tlh is a marker of V. parahaemolyticus species, so should be also present whenever one or both virulence markers (tdh and trh) are identified.
L123: the names of the proteins encoded by aerA and ahha need to be given, the genes need to be in italics
L132: Why have the strains confirmed biochemically not been chosen?
L138 -142: needs to be rephrased
L166-167: do you mean wild fish?
L170-171: meaning unclear
L201-203: needs rephrasing
L204: why talking about oysters? surely the situation is not comparable
L208-209: “this is… expectations”: ?? meaning?
L211-212: needs more explanation
L232: need full name for RBCs
L235-237: “classified as virulent… virulence genes”: ?? doesn’t make sense
L251-252: “poor storage in ponds” what does it mean?
L275: “biochemical identification” what’s the difference with the biochemical examination in 4.2.5?
L292-293: “the results were applied according to” needs rephrasing
L305: “The PCR thermal 305 conditions are illustrated in Table 2”: ?? Table 2 is the antibiotic susceptibility
Table 6: the primer sets for V. parahaemolyticus refer to reference 90, however the primers for tdh and trh and the product sizes don’t correspond to the primers mentioned in this reference.
General comments:
The background is not sufficient: more information should be given on the farm, the size of the farm, volumes of fish, husbandry and hygiene practices, etc., is this farm representative of other fish farms in Egypt?
More data should be given in the methodology/results section regarding the fish mortality: is it recurrent, is it a one -off episode? What is the mortality rate?were the 250 fish collected at the same time? The environmetal factors should be tested (temperature, salinity, turbidity, etc) => is there some variability associated with the mortality?
Discussion: the number of isolates with virulence genes compared to the total number of isolates should be discussed, the limnitations of the methods should also be discussed.
More insight should be given regarding the antibiotic resistance results and the risks for human consumption.
Author Response

(The authors gave the same response as above.)

Round 2
Reviewer 2 Report
Only comments are given here to the remaining issues
Response 1: “Response 1: Initially, the microbe was identified according to media specifications, and then the biochemical tests were confirmed. As example
Detection of the presence of Vibrio in a food sample is usually performed using selective media such as Thiosulphate Citrate Bile Salts Sucrose (TCBS) media, and then confirmed by biochemical tests”.
This is for Vibrio spp., you can’t identify V. parahaemolyticus based on colony morphology and aspect only, biochemical tests are necessary. Therefore mentioning the number of presumptive colonies is of no use and should not be mentioned. Only the number of confirmed V. parahemolyticus is needed.
These presumptive colonies can be other Vibrio or even other bacterial genus, and these numbers are of no use for the reader. Prevalence of confirmed bacteria are the main data.
Response 2: the syntax is poor and these sentences should be reworded.
L26: MAR full name should be given here
L 35: “molecular identification should be included in seafood examination in addition to other bacteriological methods” => for what purpose? Fish health and mortality investigations? Or fish food safety for the human consumer?
L58: “in both raw and wild” => raw???
Response 7: “Because the aim of the research is to detect the bacteria that cause fish contamination” => the aim of your research is maybe focused on bacteria, but it is well known that viral diseases cause also important losses in fish aquaculture.
On what basis can you say that the key cause of infectious diseases is bacteria? This is unsubstantiated.
Response 8: => The syntax of the sentence is still incorrect.
Response 11: L63: “the gene encoding” => should be genes
Response 10: “This means that each gene causes a specific symptom due to its virulence”
=> Since when genes are causing symptoms??
For the human disease, both haemolysins encoded by tdh and trh cause similar symptoms in humans. Again your sentence ("there is a variation in altered ... virulence genes") doesn’t mean anything, this sentence should be corrected.
L67: should be rephrased with: “Most Vp found in clinical cases carry tdh or trh genes”.
Response 14: the term “insensitive” is not appropriate here.
Culture followed by biochemical confirmation will only give identification of Vp.
Identification of the pathogenicity character can only be done using PCR.
Response 18: This sentence is still insufficient. You don’t describe a 16S rRNA region. It should be something like "16S rRNA is a suitable marker for…"
Besides, some recent papers mention that 16S rRNA is not the best factor for the identification of Aeromonas species as these sequences exibit relatively low discrimination.
Response 22: The phrase still doesn’t make sense: there is no relation between virulence factors and antibiotic resistance here.
Response 24: L267-276 We added this information in section 4.1
=> this is still unsatisfactory: the information you are giving in 4.1 should be part of the results and not of materials and methods.
There should be a logical progression: when conducting an investigation, you start by considering the farm and the extent of the problem with the number of fish, morbidity and mortality rate, then you examine a certain number of these fish, you conduct the post-mortem and then analyse samples.
Farm information: “in 4 sections each section consists of 13 acre (5.2 Hectares) , so the total water area production equal about 680 acre (acre = 272 Hectares)” => 13 x 4 ≠680
“the volume of production of this farm about 1400 tons / year.” => verb is missing
Environmental factors: what do these numbers mean? Are they within the limits of what is normally accepted?
Response 26: You don’t respond properly to the comment => there is no value in indicating the prevalence of presumed Vp. and A. hydrophila, only the confirmed isolates should be considered. (check any other publication on any other pathogen, data relate to confirmed microorganisms and not just presumed, not just on the number of colonies on a plate).
Response 27: text has been corrected for the tlh gene, but not for the second point: “What about strains containing only tdh or only trh”?
Very often Vp strains only contain one of the virulence genes, tdh or trh, but these strains are also pathogenic and can cause disease, so they are also important to consider.
Response 28: The prevalence rate should be the number of pathogenic Vp isolated and confirmed in relation with the number of fish being examined.
The first prevalence rates given (54% and 45%) which are for the presumptive colonies, are not useful and informative, because among these presumptive colonies there are other species than Vp and Ah, and are probably not linked to the observed disease and mortalities.
The second rates given are only rates of confirmed colonies versus presumed colonies, these rates are not informative as it is well known that media such as TCBS are not very specific.
Response 29: “The test was done, and all bands related to the gene appeared in Figure 3”
Yes, now this seems correct, however, the sizes of the bands are now different: 337 bp vs 269 bp for tdh, and 148 bp vs 500 bp for trh. This means that you have changed your PCR protocol, especially the primers. Please explain.
And how do you explain that you have the same numbers of Vp strains positive for tdh and trh despite different PCRs?
Response 31: Still needs correction => use plural, “were”, not “was”, and add “demonstrated” before high resistance.
Response 34: your explanation is good but you haven’t corrected the manuscript, please replace “considered to originate from risky sources of bacterial contamination” with your explanation directly in the text.
L218-219 : to correct “respectively isolates …respectively”.
Response 38: response insufficient, yes other virulence factors exist (other than tdh and trh) but not all V. parahaemolyticus are pathogenic, so the phrase “V.p remains pathogenic even…” is incorrect.
Response 41: this information is very interesting, and a summary should be included in the text, as it indicates the level of stress supported by the farmed fish.
Response 42: inadequate response => L302 – 304: you can’t identify at the species level with a gram stain.
Response 45: this is the same reference you gave in the first manuscript, but now you have modified your table.
Why are the PCR conditions different from this reference?
Response 46: You haven't responded to this question: "More data should be given in the methodology/results section regarding the fish mortality: is it recurrent, is it a one -off episode? were the 250 fish collected at the same time?" why did you study this particular fish farm?
Regarding the additional information you provided, are the fish from different ponds? the same pond?
Author Response
On behalf of the coauthors, I would like to thank the reviewers for their feedback and constructive comments.
Please see the attachment
